# The Diagnostics of the Condition and Management of Large-Panel Buildings Using Point Clouds and Building Information Modelling (BIM)

**Maciej Wardach** [1,*], **Joanna Agnieszka Pawłowicz** [2], **Marta Kosior-Kazberuk** [1] and **Janusz Ryszard Krentowski** [1]

1    Department of Building Structures and Structural Mechanics, Faculty of Civil Engineering and Environmental Sciences, Bialystok University of Technology, Wiejska St. 45e, 15-351 Bialystok, Poland; m.kosior@pb.edu.pl (M.K.-K.); j.krentowski@pb.edu.pl (J.R.K.)

2    Department of Building Engineering, Institute of Geodesy and Civil Engineering, Faculty of Geoengineering, University of Warmia and Mazury in Olsztyn, Heweliusza St. 4, 10-724 Olsztyn, Poland; jopaw@uwm.edu.pl

*    Correspondence: maciej.wardach@sd.pb.edu.pl

**Abstract:** Technological developments involving the implementation of modern measuring equipment and the digitalisation of civil engineering can contribute to extending the service life of buildings. Large-panel buildings constitute a large housing stock throughout Europe. This paper presents the possibility of using laser scanning to identify typical assembly defects in large-panel buildings. Based on point cloud data, numerical models were created to assess the impact of improper assembly on the elements' performance. It was indicated that using scanning to identify and monitor the displacement of structural elements does not relieve experts of the need to perform other tests. Analyses related to the possibility of using Building Information Modeling technology to manage large-panel buildings were also conducted. A parametric model was made, from which a number of possibilities of its use at every stage of the building's life were presented in an example. It was highlighted that parametric models of large-panel buildings, due to their repeatable geometry, can be copied for use in managing entire neighbourhoods. Limitations associated with implementing BIM technology in practice were also formulated. The analyses and research performed confirm the validity of implementing modern research methods in engineering practice and digitising the documentation of large-panel buildings.

**Keywords:** large-panel; BIM; laser scanning; point cloud; structural diagnostics; facility management

## 1. Introduction

The digitalisation of the civil engineering industry through the implementation of modern design and measurement methods represents a milestone in the development of this sector. Standard drafting methods were eliminated many years ago by CAD (Computer-aided Design) drawing programs. Currently, more and more projects are being created using parametric modelling, where the model has both geometry and structure containing various information. This virtual documentation contains, in addition to the dimensions of the components, material data, information about the manufacturing technology, attached product data sheets and data from manufacturing companies. In the case of existing facilities, modelling involves mapping the actual geometry in a computer program and giving the model elements all the required data, allowing the necessary information to be read freely. Implementing the actual geometry into a virtual model on the basis of traditional surveying is extremely time-consuming. Therefore, modern measuring equipment in the form of laser scanners is used. The laser beam reflected from the individual surfaces of the scanned elements creates a so-called "point cloud". On the basis of the acquired data, it is possible to create the geometry of a model of the existing object.

The authors aimed to analyse the suitability of BIM (Building Information Modeling) technology and laser scans for assessing the structural condition and management of large-panel buildings. Facilities were built most extensively in the second half of the twentieth century; hence their service life is now several decades. A large proportion of the buildings are systematically renovated and thermomodernised [1–3]. An example of a large-panel building before and after modernisation is shown in Figure 1. However, there are also poorly managed structures where the degradation processes progress faster due to the impact of an aggressive environment [4] and improper use. Degradation can also occur as a result of improperly executed refurbishments, e.g., in the form of excessive thicknesses of concrete screeds [5]. Due to the long service life, it is important to monitor the condition of these structures in order to make appropriate decisions on the need for strengthening or repair works. Analyses related to the assessment of buildings have been extensively described in [6–8]. Examples of tests on large-panel structures using a range of destructive and non-destructive methods were presented in [9]. In the case of large-panel structures, the most sensitive structural locations are the joints of prefabricated elements, the description of tests and examples of analysis of which were presented in [10]. A large number of defects also relate to facades, the studies of which have been presented in [11,12]. The paper [13] describes the anticipated directions of modernisation of large-panel buildings and characterises the possibilities of their recycling, which is particularly important in the aspect of the transition to a closed-cycle economy.

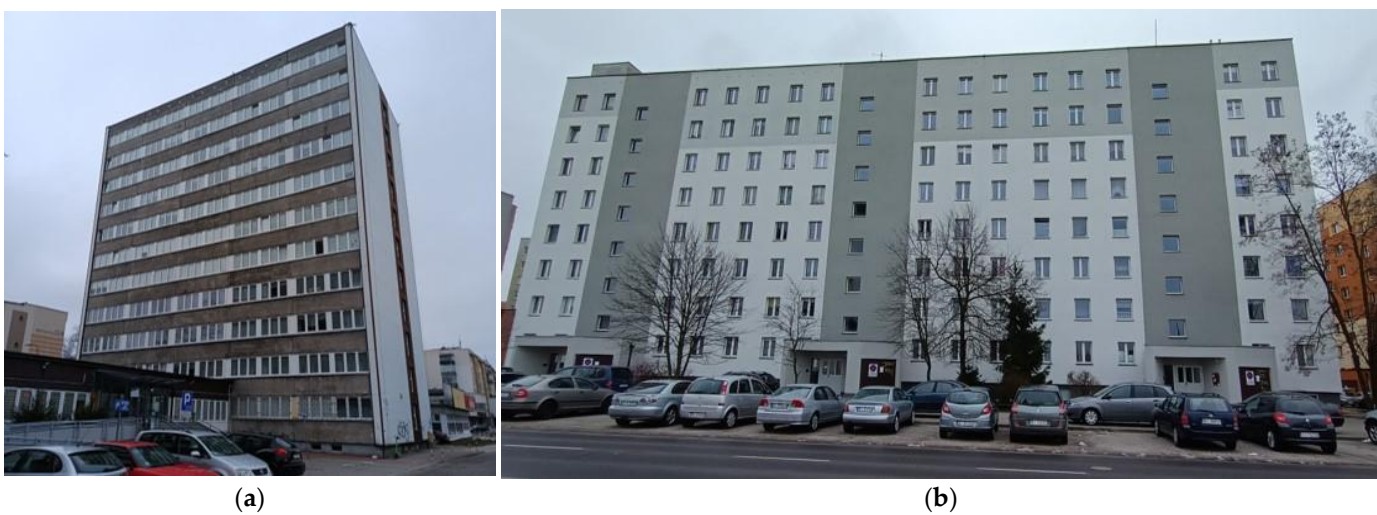

(**a**)　　　　　　　　　　　　　　　　　　　　　(**b**)

**Figure 1.** Examples of large-panel buildings: (**a**) building before thermomodernisation of the front elevation; (**b**) building after thermomodernisation.

In practice, BIM technology is most often used for newly designed buildings, where the model is used at every stage of the object's life—from conception through design, construction and operation. For existing facilities, models are mainly made for industrial or heritage (monumental) buildings.

The so-called 'research gap' was identified as a result of an analysis of the issue based on a literature review. Large-panel buildings represent a significant housing stock throughout Europe. The number of buildings makes their management and regular structural condition assessments labour-intensive. The design documentation of these buildings is still in paper form, and in a large number of cases, it is completely missing. The undertaken research topic concerns the methodology and practical implementation of modern tools to improve the management of entire housing estates and to monitor structural degradation and assess its durability. The research question relates to the possibility of using new technologies in this sector of civil engineering, which was focused on mass production, affecting the quality of construction.

This paper presents the benefits of creating models for existing large-panel buildings. This is intended as a contribution to the digitalisation of the prefabricated system building sector. The use of parametric models to manage several decades-old residential buildings is not yet observed in practice. The article aims to show how the use of a parametric model can improve the management not only of a single building but also of entire housing estates. This is due to the repetitive character of the used buildings, where most of the modelling work can be largely copied, significantly reducing the workload.

In engineering practice, condition assessments of large-panel buildings are mainly carried out on the basis of visual surveys, which can be very time-consuming and subject to high uncertainty. Given the large number of buildings, it makes sense to implement modern survey techniques to improve the entire diagnostic process. To this aim, the possibilities of using laser scanning to identify typical defects in large-panel buildings were presented, and the limitations of this method in the case of exploited buildings were pointed out. These defects are specific to prefabricated system buildings and are rare or non-existent in structures of other construction. Demonstrating the possibility of scanning for their identification and then assessing the impact of degradation on the durability of the structure may contribute to the implementation of laser scanning as a common measurement method for assessing the condition of large-panel buildings.

For this study, an unfinished building constructed in a regional OWT system and a partially exploited building erected using the same technology were selected. The acronym OWT comes from the first letters of the Polish words meaning 'Economical-Large-panel-Typical' (Oszczędnościowy-Wielkopłytowy-Typowy—in Polish). This system is characterised by a design grid with the largest module size of 5.40 m × 4.80 m. The 14 cm thick floor slabs were supported on reinforced concrete walls of the same thickness and an external beam-wall. Assembly work on the unfinished building stopped in 1980 before the roof slab was completed. The building has a full basement and five floors above ground. The structure had been exposed to an aggressive environment for more than 40 years and was gradually degrading. The second building under investigation has one underground and three above-ground storeys. Part of the building is still in use. Since the building was put into use, there has not been any structural strengthening or thermal upgrading work done.

The selection of objects is justified by the occurrence of typical defects in large-panel buildings in the form of incorrect assembly. This made it possible to present examples of the use of point cloud analysis to determine defects and their impact on the work of structural elements. The lack of need to obtain consent from building users to carry out the scanning was important. The point cloud also mapped the furnishings of the dwellings. Removal of redundant items is required for surveys, which can be inconvenient for tenants.

The authors highlight that the same measurement methods using a point cloud can be performed on in-service facilities. Monitoring of floor slab deflections can also be carried out in rooms with finish layers. Measurements must be taken at appropriate intervals to monitor the deformation of the structure. However, the caveat is that laser scanning of the facades of buildings after thermal retrofitting is not justified. The new layers of insulation and facade cover the prefabricated elements. This makes it impossible to measure displacements from the outside and to detect leaks and other types of damage. In Europe, in geographical areas where cold winters are rare, buildings with an unchanged original facade still represent a significant proportion of all large-panel buildings.

## 2. Literature Survey

### 2.1. Laser Scanning

Developments in technology are replacing time-consuming surveying with theodolites and tacheometers in favour of 3D laser scanning. This method is based on the creation of a point cloud and has a number of applications in the field of architecture, engineering and construction (AEC) [14,15]. This state-of-the-art measuring technology is particularly useful for object inventory and the identification of damage to individual structural elements. Unmanned aerial vehicles (UAVs) are being used for building facade damage detection [16].

Identification of minor damage, such as cracking, can be difficult due to background noise and different facade materials, requiring the development of suitable learning methods for identifying such degradations [17]. The paper [18] also presents the possibility of using UAVs to generate maps of thermal leaks in elevations that result in reduced energy efficiency of buildings.

In infrastructure, in addition to the use of point clouds for the design of road alignments or analysis of earth masses (soil volumes), scanning can be used to monitor the degradation of forest road surfaces [19]. This measurement method is also used to monitor the condition of massive structures such as dams [20] and underground structures [21]. A paper [22] outlines the benefits of using scanning in bridge construction, which can reduce construction time.

Numerous studies have been carried out on the improvement of point cloud processing methods and measuring equipment [23]. Current algorithms allow the automatic recognition of concrete and reinforcing bars and sleeves embedded in precast elements [24]. A paper [25] presents a method that allows the software to automatically extract individual structural elements (walls, floors, columns and beams), which can be used for quality control of construction work. Solutions are also being developed to combine 3D point clouds and 2D images to detect cracks in concrete elements [26]. Scanning can also be performed underwater, as presented in the paper [27]. Monitoring small displacements requires the development of algorithms to compare the displacement of points between adjacent cloud segments [28].

Currently, point-cloud-based modelling is widely used for assessing the condition of historic buildings [29,30] and preserving their cultural heritage [31]. Scans can also be used in industrial facilities [32]. This is particularly useful in terms of planning expansion. The large number of technological installations generates the risk of design and execution errors, which may risk manufacturing downtime. The obtained point cloud makes it possible to avoid collisions of designed installations with existing technology. Automated methods for tracking MEP (mechanical, electrical, plumbing) components during the construction process are also the subject of research [33].

Laser scanning is not only used in AEC but also in other fields, such as forestry and geology. It allows for the monitoring of tree heights [34] and the entire forest structure [35]. The paper [36] also presents the possibilities of point cloud analysis of rock masses to determine its crack network.

As a non-contact measurement technique, laser scanning is subject to a risk of errors. Measurement accuracy depends on a number of factors, such as the type of device, type of material and scanning settings [37]. It is worth noting that there are no standardised tests currently available to measure the influence of individual factors on the accuracy of the results obtained [38]. In the paper [39], a study related to the effects of different angles of incidence and distance on the measurement of concrete slabs was presented. Research related to the accuracy of crack width measurements depending on the position of the test stand was presented in [40]. The results obtained, as well as the scanning speed, are also influenced by the scanning resolution, as described in [41].

### 2.2. BIM

BIM technology can be used at every stage of investment implementation and is aimed at optimizing information transfer processes [42,43]. In the first stage, i.e., in the conceptual phase, it enables the development of solutions generating benefits for investors [44]. The visualisation and simulation of design solutions help to select the most optimal variant for stakeholders. At the design stage, the creation of inter-branch parametric models allows for minimizing the risk of errors [45].

During construction management, BIM allows for better management and control of construction. Any files can be displayed on mobile devices, and, in addition, they are constantly updated, making it impossible to work on outdated documentation [46]. By seeing visualisations, they can more easily produce complicated reinforcement or joint

details. There are many examples in the literature of large projects implemented using BIM technology [47,48]. The analyses presented in [49] indicate that it is recommended to introduce BIM education at universities to improve the learning of construction management. In the case of steel structures, BIM technology is also used in the fabrication phase, where 3D information can be transformed into data retrieved by the machines used to process the components [50]. BIM information technologies are also finding applications in relatively unusual areas, for example, to eliminate damage to large-panel buildings caused by warfare [51].

The BIM models are also used for the extensive management of existing facilities [52–54]. In order to be used effectively for building management, the model must be correctly executed, i.e., contain all the necessary information [55]. New solutions to improve the processes of updating and maintaining the models are being extensively studied [56]. The paper [57] presents interesting analyses dedicated to minimising the modelling efforts while keeping the model highly functional for facility management.

The policy of many countries to reduce their carbon footprint also determines the performance to analyse the validity of using parametric models to develop solutions that will reduce greenhouse gas emissions [58,59]. The paper [60] presents the possibility of using this technology to assess the actual global warming potential in the building design process.

Closely linked to BIM technology is the concept of the 'digital twin', which is a model of an existing building in virtual space together with real-time data processing for constant updating of the mapping. The data needed for the update is provided through the Internet of Things (IoT) [61–63]. Devices in the building must be provided with the appropriate instrumentation to allow wireless connection. An example of obtaining updates from cameras installed in a facility was presented in [64]. Researchers are using digital equivalents of existing buildings to analyse retrofit solutions to reduce the energy consumption of facilities [65]. Digital twins may find application in infrastructure developments such as highways, where a model is made using map data [66]. Research is also underway into the suitability of the model for bridge management [67,68].

In tandem with BIM, virtual reality (VR) technology is also being used in civil engineering [69]. Using virtual reality, demonstrations of designed facilities can take place, which is currently a highly requested marketing ploy. The paper [70] presents an example of using these technologies to tour a university campus and simulate an escape during a fire. The research presented in [71] also shows that the combination of BIM and VR can be used in training courses for AEC professionals. Modern technologies allow the combination of the real and virtual worlds, which may, over time, become a common practice in civil engineering.

## 3. Laser Scanning

### 3.1. Background

This section of the paper presents the potential of laser scanning to identify defects specific to systemic prefabricated buildings. These defects are relatively repetitive and occur in a wide range of large-panel buildings. The number of buildings for which there is a need to identify degradation is very high, so it makes sense to use fast and efficient measurement techniques. Properly conducted surveys allow appropriate decisions to be made regarding the carrying out of repair works and provide information to select the best solution for upgrading the facility to improve its functional properties.

Large-panel buildings have a specific construction, where the quality of the connections is an important parameter affecting the global condition of the object. However, the welded joints of the prefabricated elements are monolithised by a layer of concrete. As a result, the condition of the building is determined by the condition of the individual elements, but above all, the condition of the connections. Defective joints can result in an unsignalled failure. Analysis of the actual condition of the connections requires the use of specialised equipment, which is not easily available at the level of the practising surveyor [10]. However, the assessment of the durability of a building can firstly be based

on the evaluation of its deformation, where laser scanning is applicable. If the results do not indicate significant displacements of individual elements and no considerable damage, such as wide cracks, is found on the basis of visual assessment, the safety of the structure is not at risk. If excessive displacements are found, a more extensive analysis using specialised testing equipment is required.

The scanners differ in various technical parameters such as beam width and divergence, angular resolution, scanning speed and distance measurement error. A measuring device with a scanning speed of up to 2,000,000 points per second and a distance measurement error of $\pm 1$ mm was used to scan the large-panel buildings under investigation. The scanner can operate in a temperature range of $-20$ to $55\ ^\circ$C, and the maximum scanning distance reaches 70 m. The wavelength is specified by the producer as 1550 nm, the beam divergence is 0.3 mrad, and its diameter at the exit is 2.12 mm. The declared measurement stopper size is $0.009^\circ$, and the maximum vertical scanning speed is 97 Hz. This paper does not analyse the impact of the location of the measuring station on the results. The focus was on the possibility of using scanning to identify the degradation of large-panel buildings. Based on the tests carried out, defects resulting from improper assembly and aggressive environmental influences were identified.

The identification of degradation was based on point cloud analysis using an open-source software named CloudCompare v2.12.4 [72]. It is used to process 3D point clouds and is equipped with algorithms that allow, among other things, resampling, colour handling, interactive segmentation and normal field support.

### 3.2. Facade Surveys

The laser scanning consisted in making a point cloud of the western part of the surveyed, unfinished building. The measurement was taken from three positions. The data obtained were combined into a single point cloud to analyse the correct installation of the wall elements. A fragment of the gable wall was obscured by a tree, which prevented the laser beam from reflecting off the prefabricated panels. This is one of the limitations of such survey methods, which can make it difficult to work in heavily wooded areas. The scanning device and sections of the point cloud are shown in Figure 2.

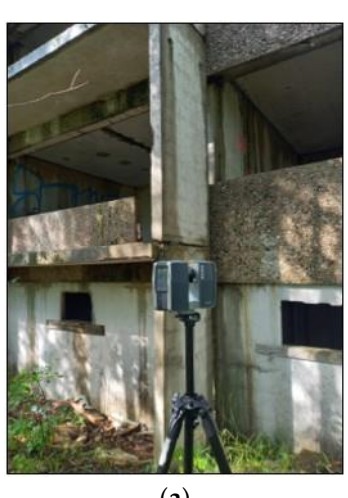 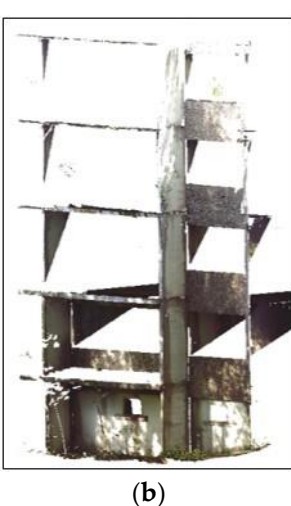 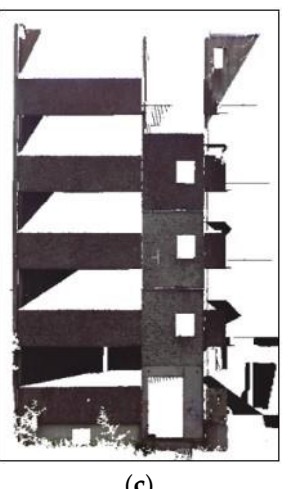 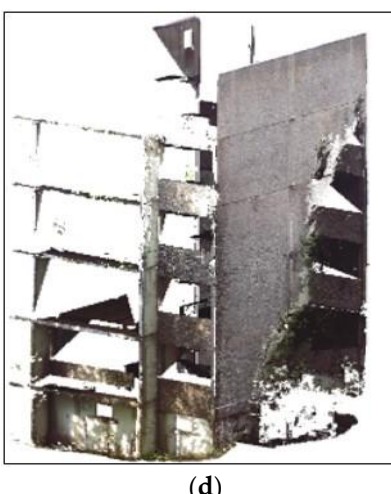

(**a**)        (**b**)        (**c**)        (**d**)

**Figure 2.** Ground-based laser scanning: (**a**) measuring device; (**b**) back elevation point cloud; (**c**) front elevation point cloud; (**d**) joined point cloud with gable wall view.

The gable walls of the large-panel buildings in the regional OWT system were made as prefabricated, three-layer walls. The inner layer was a 14 cm thick reinforced concrete bearing wall, followed by a 5–6 cm thick polystyrene insulation layer. The facade layer was made of 5–6 cm thick reinforced concrete, which had an architectural function. The critical points of the facade were the horizontal and vertical joints of the panels. These

joints, in the case of improper installation or inadequate sealing, are places through which moisture penetrates the building [12]. This degrades the structural elements and can lead to the growth of fungi and mould in the flats. According to the design guidelines, the external vertical contact should have a width of no more than 20 mm. The joints along their entire length should be sealed with sealing putty, and the insulation layer was to be protected from the inside with bitumen felt. A steel hood with aluminium sheeting was also supposed to protect against water penetration. The detail of the joint sealing is shown in Figure 3a.

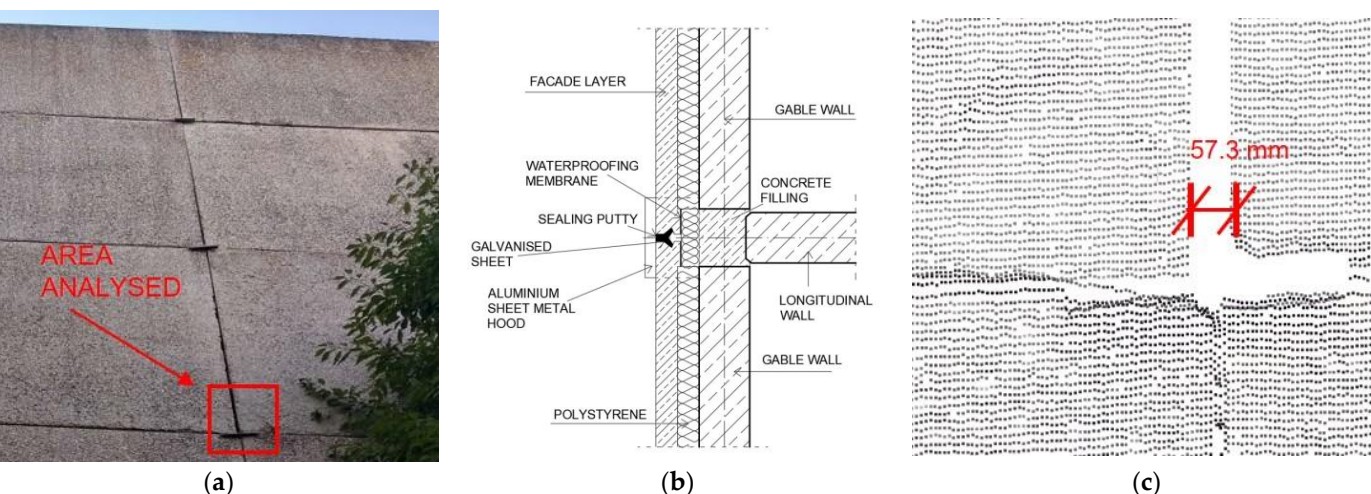

**Figure 3.** Identification of gable wall defects: (**a**) view of scanned gable wall; (**b**) sealing detail; (**c**) measurement of the gap between elements made on the point cloud.

On the basis of the point cloud analysis, an incorrect installation of the facade panels was identified. The defect was located by visual assessment using laser scanner images. The vertical distances between the elements were irregular and excessively open in places, reaching a width of more than 55 mm compared to the designed 20 mm (Figure 3b,c). Localised spalling of the texture layer was also located, which is most likely the result of damage during incorrect transport or installation.

### 3.3. Vertical Joint Deformation Surveys

Non-verticality of structural elements can result in incorrect operation of structural components due to the redistribution of forces that is different from that assumed in the design. An example of incorrect assembly, which can occur in large-panel construction, is the setting of wall slabs at unintended eccentricities. For instance, in the analysed building, a wall whose eccentricity in relation to the wall of the lower storey was 15 mm. This is a value that, according to current standards [73], is considered to be the limiting value for the investigated case. In order to analyse the effect of the eccentrically applied load on the stress distribution in the wall of the lower storey, a 3D calculation model was made. The point cloud, together with the analytical model, is shown in Figure 4.

Numerical calculations were carried out in ANSYS 2021 R1 [74]. To simplify the analysis, a single, two-storey module was modelled from the analysed building. In order to analyse the effect of the eccentricity on the stresses in the wall of the lower storey, the calculations were carried out in two schemes. In the first scheme, the walls were modelled without eccentricity. In scheme two, the wall of the upper storey was placed on a 15 mm eccentric in the internal direction, which was in accordance with the measurements obtained by laser scanning.

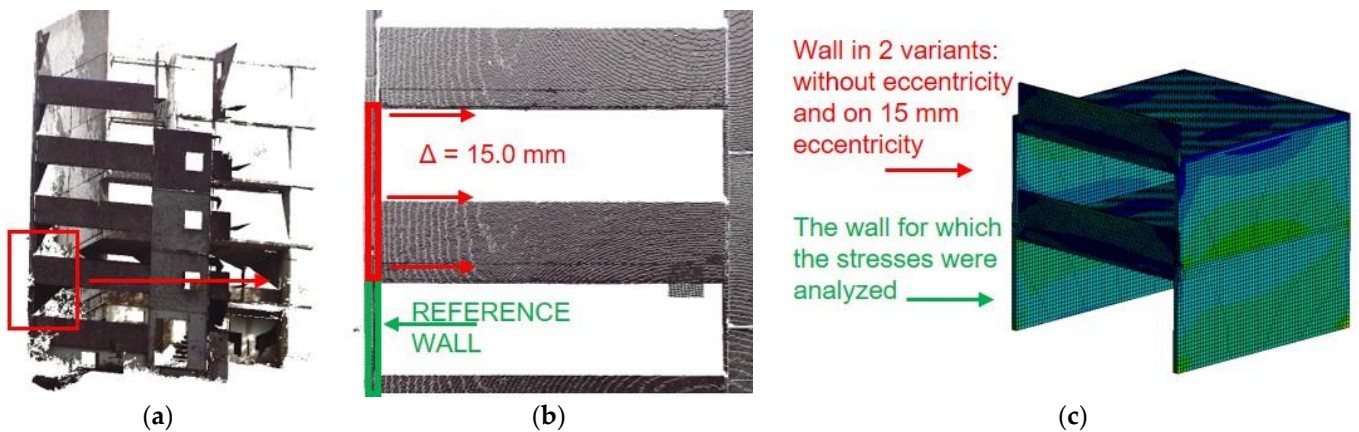

**Figure 4.** Identification of wall eccentricities: (**a**) point cloud view with the analysed area marked; (**b**) identified wall eccentricity; (**c**) computational model.

The walls were modelled as 14 cm thick C16/20 concrete with dimensions of 4.80 m × 2.52 m and 5.40 m × 2.52 m. On the walls were supported the floor slabs and beam-walls, which carry the loads from the floors and form the front wall of the building. Material stress–strain curves were entered into the program in accordance with EC2-1 [75]. The walls were loaded with pressure acting on the upper surfaces to reflect the transfer of loads from the upper storeys. The slabs were also loaded with pressure applied to the entire floor surface. Load values were assumed for a completed, in-service building according to EC1-1 [76]. The imposed loads were assumed to be 2 kN/m$^2$, variable loads from partition walls were assumed to be 1.5 kN/m$^2$, and dead loads were assumed to be 2 kN/m$^2$. The self-weight of the structure was also included. Load combinations were adopted according to EC0 [77].

The walls were modelled as solid using SOLID187 elements, which are defined by 20 nodes with 3 degrees of freedom. The FEA mesh was selected with a HEX20 shape (20 nodes hexahedron) with a size of no more than 100 mm, which was sufficiently accurate for the analysis under consideration.

The static scheme of large-panel structures is more complicated than that of monolithic structures. Monolithic reinforced concrete structures form a rigid whole due to the continuity of reinforcement between adjacent elements. In prefabricated construction, the elements were joined on site with steel connections and their contacts were filled with concrete. Each such joint is characterised by a different individual susceptibility. In order to simplify the analysis, the connections between the elements were assumed to be fixed in the model, and the wall supports were assumed to be fixed, which significantly reduced the complexity of the task and the calculation time. The calculations carried out are intended to demonstrate the difference between the stresses depending on the size of the wall eccentricity and not to analyse the working of the connections; hence the accuracy obtained with the simplifications adopted was considered sufficient.

The results of stresses in the lower wall loaded without and with eccentricity are presented in Figure 5.

For a wall scheme set on an eccentric, the stresses of the lower storey wall are increased by about 9% in the upper part of the wall and by 19% in the lower part. For an eccentric that is equal to the maximum normal deviations, the increase in stress is relatively small and does not threaten the durability of the structure. In the case of a larger eccentricity, the redistribution of forces can be significantly disturbed, resulting in the failure of structural elements and threatening the durability of the wall. In addition, it should be noted that the study was carried out for a relatively low building with five overground storeys. Buildings with 11 storeys have also been constructed using the system analysed. In the case of eccentricity in the walls of tall buildings, which are highly vertically loaded, the stress increment will be proportionally higher.

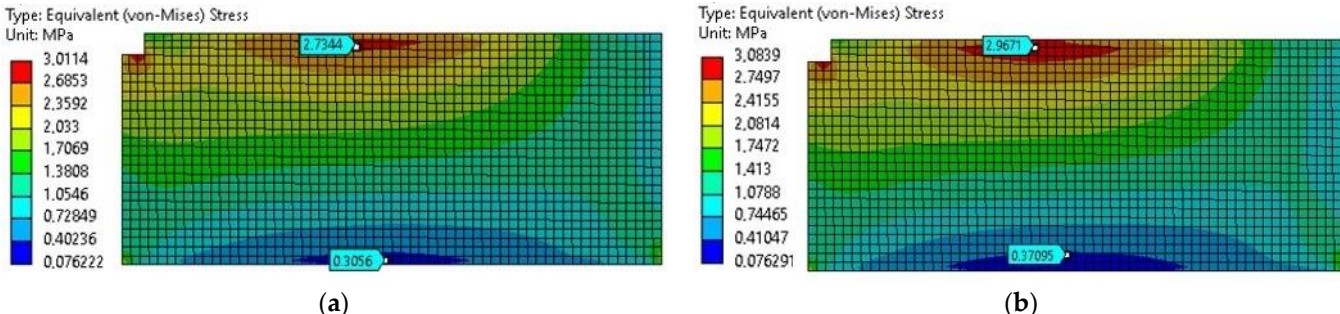

**Figure 5.** Stresses in the wall of the lower storey: (**a**) stresses in the case of the wall of the higher storey installed without an eccentric; (**b**) stresses in the case of the wall of the higher storey installed with a 15 mm eccentric.

### 3.4. Floor Slabs Deflection Surveys

In the system analysed, the floors were made as slabs with dimensions of 2.70 m × 4.80 m, which were supported on three edges: on two walls and an external beam-wall. The free edge was the connection between adjacent floor slabs, which were filled with concrete and reinforced with a spiral wire. This was to protect the slabs from uneven deflection and to ensure that the slabs cooperated properly in transferring loads. The conditions for supporting the slabs in this way ensured very little deflection with a small section height of 14 cm.

Using one measuring station, the deflections of the slabs above the ground floor in 2 rooms were measured. The deflection results are presented in Figure 6a. There were no finish layers in the unfinished building, so the floors were only loaded by their own weight and exposed to the weather. Numerical calculations were carried out to compare the actual and theoretical deflections.

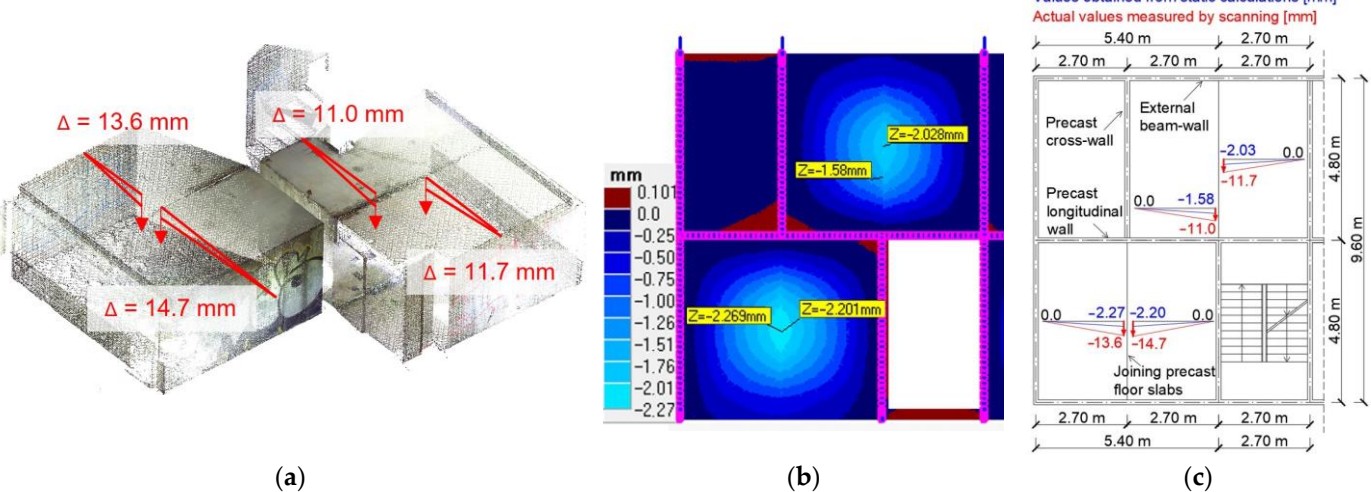

**Figure 6.** Verification of slab displacements above the ground floor: (**a**) reading of deflections from point cloud; (**b**) deflections from dead weight based on numerical calculations; (**c**) comparison of actual deflection values with the theoretical values.

The calculations were made with the commercial calculation program ABC slab [78]. The 14 cm thick floors were supported on walls also 14 cm thick and on an external beam-wall. The walls were modelled as linear supports taking into account their vertical susceptibility. The calculations were based on concrete parameters C16/20. The quantity and cross-sections of the reinforcement were determined on the basis of the archived design documentation. The calculation algorithms were based on the assumptions of the classical thin plate theory. The calculation of deflections of the cracked floor slabs was carried

out iteratively. Successive iterations consisted in modifying the stiffness according to the degree of cracking and the rheological quantities. A maximum crack opening limit of $w_{k,lim}$ = 0.3 mm was assumed in the model. The results of the deflections from self-weight are shown in Figure 6b. A comparison of the results obtained from in situ measurements and numerical calculations is shown in Figure 6c.

On the basis of the performed analysis, it was determined that the actual deflection is up to 668% greater than the theoretical deflection resulting from the dead weight of the floor slabs. Such a large difference may be due to many years of exposure of the floor slabs to aggressive external environmental factors causing degradation and the occurrence of rheological processes in the form of concrete creep. The excessive vertical displacement is also due to the fact that the adjacent floor slabs have not been properly joined, from which the concrete mixture connecting the two elements has been washed out due to moisture penetration. The cyclical interaction of extremely low and high temperatures may also have caused the horizontal displacement of the slabs at the supports due to the thermal expansion of the concrete and steel, increasing the rotational susceptibility of the horizontal joints.

This high initial deflection makes it difficult to carry out specialised repair work. The slabs, when permanent and variable loads have been added, will receive a multiple of the self-weight load, which will further increase the displacement values. Excessive displacements cause, in turn, many partition wall failures in the form of excessive cracking and even "slump" phenomena [5].

## 4. Parametric Models of Large-Panel Buildings—Examples of Application

### 4.1. Description of the Subject of the Analysis

In the large majority of cases, facility management using a BIM model is limited to facilities that were designed using this technology, and the model was already present from the concept stage. In practice, the use of BIM for existing buildings that were not designed using the parameter model is focused on facilities of high cultural value (heritage buildings) or of great economic importance (industrial, infrastructure facilities). Given the number of large-panel structures, it makes sense to introduce solutions that can contribute to improving the efficiency of management at each stage of the life of the building.

There are several advantages for the property manager in having a model of an existing large-panel building. The most important is having all the information in several files. This data reading is much more readable than the standard search for necessary information in paper documentation, some of which, in the case of a decades-old building, may be lost or destroyed. In addition, the lack of updating the documentation makes it outdated and, therefore, useless for facility management. The efficient workflow is another benefit of using BIM technology. By working on a model connected to an integrated digital platform, when data are inputted, e.g., concerning the occurrence of a defect, information to all stakeholders appears immediately (e.g., in the form of email notifications). Traditional methods are mostly based on emails sent independently by the responsible person. This involves risk factors such as delays that may result, for example, from absence from work and the possibility that one of the stakeholders may be overlooked.

The use of BIM technology should be compatible with the achievement of technical efficiency and the realisation of economic value. Otherwise, it may not bring added value to a project [79]. Important in relation to the efficiency of the use of parametric models is the integrity of the software. Without integrity, the correct sharing of information is very difficult [80]. The use of laser scanning and making parametric models on this basis is also associated with high initial investment costs. Analyses carried out in [81] indicated that the introduction of scanning and BIM technology is economically efficient, especially for large-scale projects. Undoubtedly, large-panel housing estates, due to the number of buildings, belong to these.

In order to determine the suitability of parametric models for the management of large-panel buildings, the authors conducted extensive research into BIM technology and systemic prefabricated buildings. The research included literature studies, tracking of

current trends in the civil engineering sector, industry insights from construction experts and a large amount of our own research on systemic prefabricated buildings [4,5,9,10]. The authors verified the results of their research carried out using various methods. The verification was carried out on the example of detailed condition analyses of more than 100 buildings located across the country in 6 out of 16 provinces in Poland. For example, in the province of Mazovia, more than 100 buildings were examined [3,12]. The subsequent research presented in this article contributes to the development of a research methodology for large-panel buildings. A research methodology related to the feasibility of using BIM technology in large-panel facilities is presented in Figure 7.

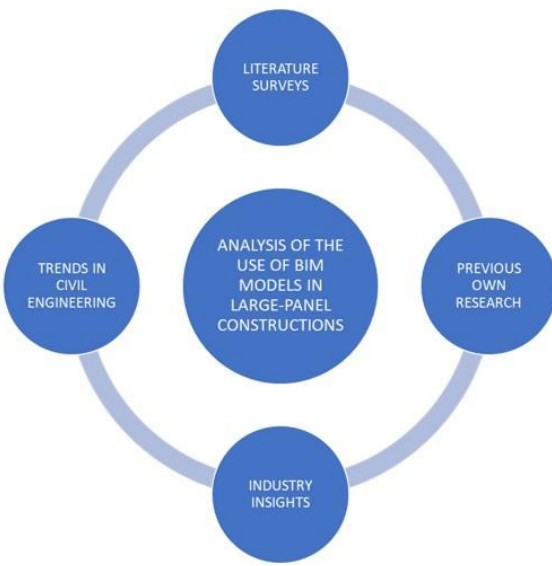

**Figure 7.** Methodology of the research carried out to identify the feasibility of using BIM models in large-panel buildings.

The analyses carried out indicated a significant interest in BIM technology among engineers, as well as investors and facility owners. Drawing knowledge from the afore-mentioned sources, including from cases where BIM technology for facilities management has already been implemented [82,83], the authors have identified opportunities for the application of parametric models in large-panel buildings. The paper presents benefits that are not limited to individual facilities but can be implemented to manage entire housing estates. This is due to the very repetitive architecture of buildings located within the same housing estates. This makes the work of digitising the documentation easier. The copying of geometry significantly reduces the labour intensity associated with modelling. The conducted analyses were carried out using a sample parametric model made by the authors. The partially decommissioned large-panel building surveyed, together with the parametric model, is shown in Figure 8.

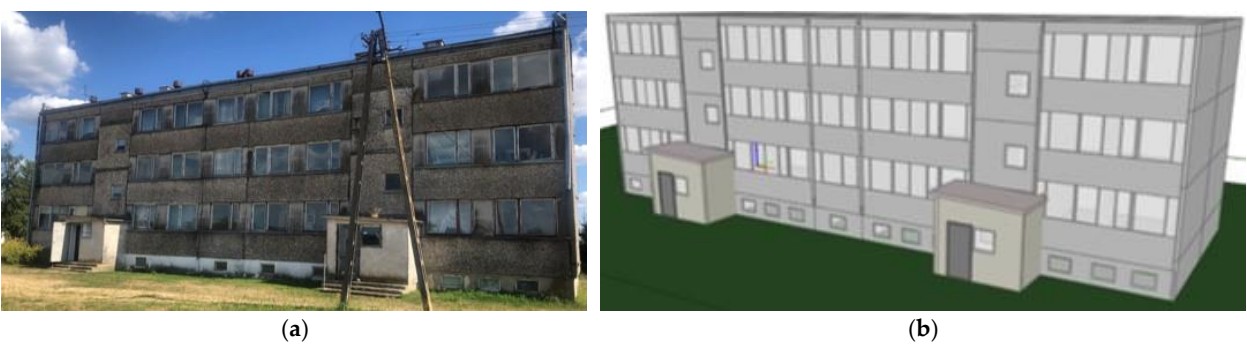

(**a**)　　　　　　　　　　　　　　　　　　　　　　　(**b**)

**Figure 8.** Digital version of existing large-panel building: (**a**) existing building; (**b**) BIM model.

*4.2. Analysis of the Possibility of Using Parametric Models Based on Own Research*

The authors collaborated with leading research institutions in Poland in the field of testing large-panel structures. The presented research was discussed at a wide range of expert meetings, where it was met with approval. Based on the analyses carried out, several possibilities for using the model of an existing large-panel building were identified, as shown in Figure 9.

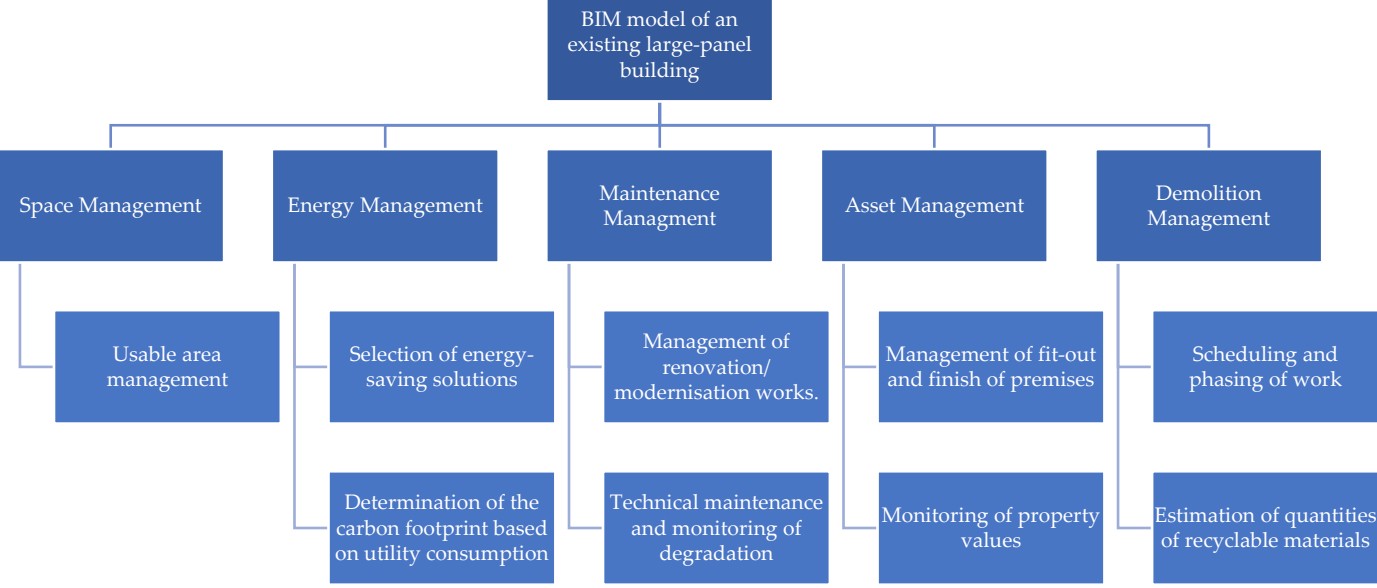

**Figure 9.** Possibilities of using a BIM model of large-panel buildings (the author's own study).

Space Management refers to the management of a facility's floor space, which can be effectively realised using a parametric model. When flats are subdivided or merged, it is possible to mark such changes on the model and read off the newly created space. This makes it easier to manage the fees charged to individual property owners. Precise floor plans can also be entered into the model, including the positioning of partition walls. This is particularly important for the management of office buildings, which were also built using large-panel building technology.

Energy Management is the management of resources used in the erection and operation of facilities. Increasingly restrictive directives are now emerging in Europe to minimise the produced carbon footprint and to specify the need for a transition to a closed-loop economy. In order to assess potential environmental risks, an environmental life cycle assessment (LCA) has been developed. The essence of this method is to estimate and assess the environmental consequences of a technological process. BIM models make it possible to carry out the most reliable LCA analysis, as they reproduce the actual geometry of buildings and minimise the risk of errors that can arise when working on 2D documentation. Using a parametric model, it is possible to calculate the carbon footprint of a building in operation based on utility consumption. In advanced models, it is also possible to estimate energy consumption depending on modernisation solutions in the form of insulation, window replacement or renovation of installations. This allows a variant analysis of the planned changes so that the most economical and environmentally friendly option can be selected.

Maintenance Management refers to the service of a facility. The use of a parametric model for this purpose offers a number of advantages. Occurring faults reported by users can be marked in the model, making it easier to organise repair works. Current algorithms allow fault notifications to be automatically sent to the relevant maintenance companies. A reminder can also be sent to persons responsible for carrying out technical inspections, minimising the risk of negligence resulting from a lack of notification of the need to carry out controls. Examples of identified facade defects in a partially decommissioned building

marked on the BIM model are presented in Figure 10. The marking of the defects, together with the assignment of photographs, allowed the selection of appropriate repair methods and enabled the creation of an enquiry to obtain a reliable cost estimate. This formed the basis for an economic analysis of the reasonableness of the repair. Currently, many buildings are undergoing thermomodernisation consisting of insulation and a new facade. The models can be used to generate a visualisation of the building after renovation work. Appropriately adopted architectural solutions allow buildings to be made more attractive, thereby increasing the value of the property.

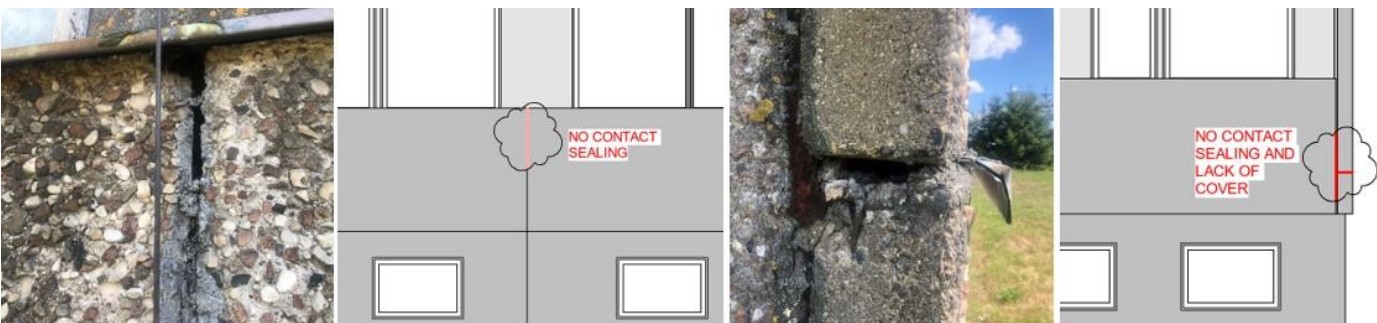

**Figure 10.** Facade defects marked on the parametric model.

Based on the model developed using the point cloud, it is possible to create numerical models taking into account the actual geometry, including the non-verticality of the elements. The analytical model makes it possible to analyse the impact of degradation on the durability of the building and, if its durability is threatened, to select appropriate repair works. The authors carried out an analysis of the possibility of opening up prefabricated internal walls in order to increase the surface area of the dwellings and thus increase their architectural potential. For this purpose, a parametric model was used, on the basis of which a numerical model was made (Figure 11a), and the necessary static calculations were carried out. Currently, many cities are also expanding their underground transport networks, i.e., subways. With the model, it is possible to determine the effect of vibrations caused by the train depot on the operation of the structure and to determine if strengthening is necessary.

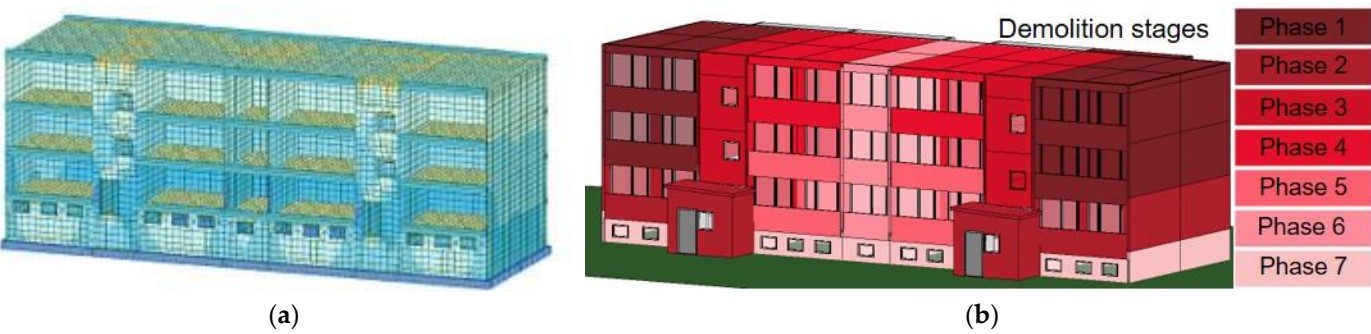

(**a**)　　　　　　　　　　　　　　　　　　　　　　　　　　　(**b**)

**Figure 11.** Examples of the use of the parametric model: (**a**) calculation model; (**b**) organisation of demolition work.

Asset Management is also a process that can be improved by using parametric models. Including information on the furnishings and finishes of the premises, together with the degree of deterioration, enables a more precise valuation. If the property manager has a building with flats for rent, this allows the value of individual units to be monitored and fees to be set in proportion to the standard.

Demolition Management is the process involved in the dismantling of buildings. If the sustainability of the structure is identified to be at risk and repair work is not reasonable, it



is necessary to demolish the building. Large-panel buildings are mostly located in densely built-up housing estates and around streets with increased traffic. This is due to the fact that the buildings were erected decades ago when cities were less developed than today. Over the years, facilities have been surrounded by more and more new road infrastructure and more facilities, which has increased the density of development. For this reason, the demolition of a building must be carried out in a way that does not threaten the safe operation of neighbouring buildings and road users. Demolition works can be planned using BIM models, in which dismantling processes can be phased, and a suitable schedule for carrying out the work can be drawn up. In addition, it is possible to estimate the number of materials that can be recycled. The authors also carried out a comparative analysis of the costs of demolishing an existing structure and constructing a new building in order to determine the viability of renovating the studied partially decommissioned building. Based on the parametric model, a phasing scheme for the dismantling work was created, as shown in Figure 11b. This enabled a schedule of works and construction cost estimates to be created. The sequence of works was chosen, taking into account the simultaneous operation of two demolition excavators located on opposite sides of the structure.

## 5. Discussion

The digitalisation of civil engineering is leading to the elimination of paper documentation from the construction process. This brings a number of benefits in terms of organising data, saving resources and storage space and making it easier to organise work. In the case of newly constructed buildings, work on the model usually starts at the concept stage. The model is then used during design works, execution works, and, in turn, by the property owner to manage the facility.

The number of large-panel buildings with a long service life is forcing more and more repairs, thermal upgrades or demolition works. The common lack of archival project documentation is also associated with a lack of basic information about the facility. The highly repetitive nature of large-panel buildings greatly simplifies modelling processes. At present, practically every newly designed building is characterized by its individual architecture, which makes it impossible to reuse a model of a building to manage another facility. The subsidence of the parametric model is characterized by a number of advantages in the management of entire large-panel estates. In this type of construction, this is particularly important as these buildings are energy inefficient, which requires numerous modernisation works to be carried out. In addition, as a result of the increasing age of use, often exceeding 50 years, consideration of demolition of these structures will become more frequent. This justifies analyses involving the dismantling and recycling of these structures.

The limitations of managing large-panel buildings are largely related to choosing the proper software and compatible applications that are easy to use for both occupants and property owners. In addition, facility owners have to expect a long time to implement the technology and its high costs.

Laser scanning can be successfully used to create the geometry that forms the basis of a parametric model. The modelling process for sanitary and electrical installations can be relatively problematic. This process for finished buildings will rely on the transfer of information from traditional paper documentation to digital form. The reason for this is that laser scanning cannot be used to read cable routes and pipes hidden in walls and floors. The most labour-intensive part of modelling is assigning all the necessary information to the model elements. Sources of data that will be implemented into the model can be archival project documentation (if available), as well as survey results and inventories using traditional survey methods, including local excavations. The most labour-intensive part of modelling is assigning all the necessary information to the model elements.

The use of laser scanning is not just limited to taking a point cloud to create a BIM model. Analysis of the resulting data allows for the analysis of structural degradation in the form of excessive displacement and the identification of damage and discontinuities. Large-panel constructions are characterised by numerous defects caused by improper

installation. This is due to construction work being carried out by unskilled workers, inadequate supervision and limited access to materials. This has resulted in walls being installed on eccentricity, facade seals not being properly made or prefabricated joints being incorrectly constructed. The high accuracy of the point cloud and the short measurement time make it possible to determine the precise geometry of the object. The acquired laser scanning data can be implemented into numerical programs, allowing the impact of excessive assembly deviations on the operation of structural elements to be analysed. Regular measurement work makes it possible to monitor the progress of degradation and assess whether strengthening work is required.

The use of laser scanning to analyse facade degradation is applicable to buildings before thermomodernisation, where prefabricated elements are not covered by additional layers. For facades obscured by vegetation, it makes sense to use drone raids instead of ground-based scanners. Drones allow scanning of the building from different positions, including avoiding trees. Another solution to the problem is to perform tests in the winter months (lack of leaves) or mechanically remove fragments of trees and vegetation. The analysis of floor deflections has to take into account the presence of finish layers and the possibility of their uneven thickness, which can lead to false results. Hence, for large-panel buildings in service, it makes sense to use laser scanning to monitor deformation growth or to take measurements after the removal of finish layers. The non-verticality of facade walls in the case of buildings before thermomodernisation can be determined from a measuring station outside the building. Measurements of internal walls are limited by the presence of finishing layers, which should be removed in order to obtain accurate results. Only monitoring of wall displacements can be carried out on prefabricated elements covered by finish layers, but it should be noted that the thickness of the layers must remain unchanged between successive measurements.

Verification of the results obtained should be carried out through a variety of test methods. In practice, there is a major limitation due to the availability of specialised testing equipment. However, results of similar validity can be obtained using different test methods. The requirement is their verification.

Structural deformation identified by scanning may be a sign that more specialised testing of structural components and their connections is required. These tests include both destructive and non-destructive testing in the form of instruments using ultrasonic, radiographic, electromagnetic or sclerometric methods.

Diagnostics of existing buildings should be carried out by experts with relevant professional experience. Buildings that have been several decades old are characterised by different states of degradation. The analysis of their durability is based on the assumption of many simplifications related to both the static scheme and the physical and strength properties of the materials. The analytical models performed in a computer program also represent simplifications assumed by the expert. Any assumption and subsequent calculation and interpretation of the results should be made by a person with significant experience in structural diagnostics. This person has the responsibility to make appropriate decisions to enable the continued safe operation of the structure.

## 6. Conclusions

Based on the conducted research, the following conclusions were drawn on the possibility of using parametric models and laser scanning for structural health diagnosis and management of large-panel buildings:

- Parametric modelling of existing large-panel buildings is useful for improving site management processes during all stages of a structure's life after its erection, i.e., operation, retrofit and demolition;
- Models of large-panel buildings, due to the repetitive nature of the geometry, can be copied to a large extent for use in the management of entire housing estates, significantly reducing the labour intensity associated with modelling;

- Laser scanning is the most efficient way of measuring the geometry of a building, which is the basis for creating a virtual model of the object;
- Point cloud analysis allows for the identification of non-verticalities, displacements of elements and facade degradation;
- Point cloud data can be implemented as assumptions for calculations in numerical programs, allowing the impact of improper assembly on the working of structural elements to be analysed;
- The use of the point cloud to identify assembly defects in operational buildings where elements are covered with finishes and buildings after thermomodernisation is limited and requires the removal of finishes or is limited to displacement monitoring.

BIM technology allows for a more efficient flow of information, which is key to effective building management. The digitisation of documentation and management processes for large-panel buildings can contribute to extending their service life as a result of monitoring the condition of the structure and carrying out analyses related to modernisation. A major constraint to the implementation of BIM technology is the need to purchase specialised software, model the objects, transfer the necessary information into digital format and train staff accordingly. This results in a lot of work and high initial costs. In order for implementation to take place, it makes sense to educate facility managers and users about the benefits of using parametric models.

The analyses carried out in the article contribute to the development of civil engineering digitisation in relation to existing large-panel buildings. Currently, the design of structures is increasingly being created using BIM technology, but in the case of existing buildings that are not heritage or industrial objects, the creation of digital models is not widely practised. Making virtual models of large-panel buildings made from a point cloud allows for efficient management of facilities at each stage of the life of the structure.

The durability of large-panel buildings is highly dependent on the joint condition of the structural elements that are covered by the concrete layer. Therefore, it is difficult to survey these areas. The use of laser scanning makes it possible to assess the deformation of the structure. When excessive displacements occur, they can indicate the degradation of structural elements and their connections. In such cases, the structural surveyor assessing the condition of the structure is obliged to carry out supplementary tests. Only by comparing the results of several testing methods is it possible to draw effective conclusions about the actual condition of the structure and to make effective decisions with regard to the further safe use of the building.

Scanning directly allows for the identification of defects typical for prefabricated construction in the form of incorrect installation of prefabricated elements or leaks in facade joints. The possibilities of using this measuring technique are, however, limited by the presence of layers covering the structural elements. In such cases, measurements are limited to monitoring the displacement increase.

Future research directions will be related to analysing the possibility of using drone (UAVs) raids to automatically identify facade defects. It is planned to use drones equipped with both scanners and thermal imaging cameras to identify sealing defects in facade panels, which may look undegraded on the surface. Due to the large number of buildings, it makes sense to develop methods that minimise human involvement in identifying degradation. Automating this process will allow partial automation of structural health monitoring.

**Author Contributions:** Conceptualisation, M.W. and J.A.P.; methodology, M.W., J.A.P., M.K.-K. and J.R.K.; software, M.W. and J.A.P.; formal analysis, M.W., M.K.-K. and J.R.K.; investigation, M.W. and J.A.P.; resources, J.A.P. and J.R.K.; writing—original draft preparation, M.W.; writing—review and editing, M.K.-K. and J.R.K.; visualisation, M.W.; supervision, M.K.-K. and J.R.K. All authors have read and agreed to the published version of the manuscript.

**Funding:** The research was carried out within the scope of work no. WI/WB-IIL/2/2021 and financed by resources from the Ministry of Education and Science of Poland.

**Data Availability Statement:** Not applicable.

**Conflicts of Interest:** The authors declare no conflict of interest.

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
