# Peer review of "The Diagnostics of the Condition and Management of Large-Panel Buildings Using Point Clouds and Building Information Modelling (BIM)"

_buildings, doi:10.3390/buildings13082089_

Round 1

Reviewer 1 Report

The novelty of this paper can be published in “Buildings”. However, the topic, research approach, and applied technology are not new. Thus, my decision is an acceptance with major revision.

Here are my comments on improving the manuscript:

1. Overall: Many studies have successfully demonstrated the proposed technology applications in building construction management. Why do the authors conduct this study again? The research contributions are weak. Please kindly explain.

2. Abstract: This section lacks research limitations and contributions. Please kindly update.

3. Introduction:

- This section is too long, causing be tired to read. Please consider separating it into the introduction and literature review sections.

- Due to a weak contribution, please address new research gaps and then emphasize why the authors do this study again.

- To clarify the introduction, please consider stating the research questions and objectives.

4. Literature review

- Besides the laser scanning review, the research lacks a thorough literature review regarding BIM applications in construction management. Please update.

- Please consider dividing this section into subsections providing topic-related analysis as mentioned above.

5. Parametric models of large-panel buildings - examples of application

- Please provide a figure to illustrate the whole research methodology.

- Please thoroughly discuss Figure 8 regarding the possibilities of using a BIM model of large-panel buildings based on fields of management, which are stated in this Figure.

- This section lacks the evaluation of construction experts on the proposed technology's effectiveness in practice. Please update.

6. Discussion

- Please consider providing a discussion to discuss significant criticism and research limitations.

7. Conclusion:

- The conclusion would be improved if the authors provide research limitations and emphasize future work. Please update.

8. References:

- Some references are too old (over ten years). Please replace the new ISI articles in order to improve this section.

Reviewer 2 Report

In this manuscript, the authors present the detail workflow for the condition evaluation and management of large-panel buildings by using point clouds collected by laser scanner and BIM models. However, the innovativeness and perspectiveness is not significant for the journal publish This manuscript is better submitted to a conference. I suggest the authors could add some automatic detection method to enhance the efficiency of the data analysis.  

Nil

Reviewer 3 Report

This manuscript presents the possibility of using point cloud collected by laser scanning to identify typical assembly defects in large-panel buildings. The results show that point cloud analysis allows the identification of non-verticalities, displacements of elements and facade degradation. The manuscript is easy to follow. I just have some minor comments:

1.Line 196: “an incorrect installation of the facade panels was identified.” The authors did not provide information on how the determination was made. It is important to know whether it was done through manual interpretation or using some automated method. This is crucial because manual interpretation is inefficient and not practical for a large number of buildings or large-scale structures.

2.In line 212, the term "a 3D calculation model" is mentioned without any relevant explanation. I did not come across any description regarding this model.

3.When conducting numerical calculations, the authors used existing software or models, which are generally based on empirical knowledge. It is important to clarify whether these models are suitable for the study area. Please provide an explanation.

4.How does the author propose to handle buildings with vegetation obstruction on their facades?

Round 2

Reviewer 1 Report

The revised paper can be published in “Buildings”. Thus, my decision is an acceptance for publication.

Author Response

Dear Reviewer,

Thank you for appreciating the corrections we made to our article. Thanks to the reviewer's earlier suggestions, the quality of the manuscript has improved.

Reviewer 2 Report

In this manuscript, the author provides a detail workflow for the defection detection workflow for the large-panel buildings by using dataset collected by terrestrial laser scanner. Here is some comments and suggestion for this work:

1.     For the facade defection detection work, images collected by UAV platform are also widely used. The authors should present them in literature review section.

2.     Please pay attention to the usage of abbreviation, please give full spelling when it is used for first time, such as BIM and OWT.

3.     The author gives too much background knowledge about the laser scanner and BIM which make the manuscript tedious. Please modify Section 2 to make the manuscript easy to read. 

4.     Line 294-297: please give the reference of this statement.

5.     The author claimed the efficient workflow is one of the most important contributions, please give the time consumption compare between proposed method and traditional methods. 

6.     The author should give some potential research directions in the future.
